# Analysis of key parameters influencing the permeability of cement sheath based on multiphysical fields

Luo Wei[1,2], Weidong Zhang[2], Kewei Xu[1], Jingwei Yang[2], Yangyang Liu[2], Wei Xiao[2], Mingji Wei[2], Liqin Qian[2,3]*, Chengyu Xia[2]*

1 Engineering Technology Research Institute, Petro China Southwest Oil and Gas Field Company, Chengdu, China, 2 Cooperative Innovation Center of Unconventional Oil and Gas, Yangtze University (Ministry of Education & Hubei Province), Wuhan, Hubei, China, 3 Clean Energy Automotive Engineering Center, Tongji University, Shanghai, China

☯ These authors contributed equally to this work.
* lqqian@tongji.edu.cn (LQ); xiachengyu2023@126.com (CX)

**Data Availability Statement:** The data can be accessed through my article, while the remaining data is not publicly available due to project confidentiality.

## Abstract

This paper develops a finite element analysis model to investigate the seepage characteristics of cement sheaths, considering the flow properties of their porous medium. The model's applicability under various conditions was evaluated through grid sensitivity tests and model validation, indicating that it effectively captures the seepage behavior of cement sheaths with a reasonable degree of reliability. Key parameters, including cement sheath length, permeability, gap structure, pressure differential, and fluid properties, were analyzed using finite element methods to determine their impact on seepage flow. The findings reveal that crack width, permeability, and cement sheath length significantly influence seepage flow in both liquid and gas media. These insights enhance the understanding and prediction of cement sheath seepage characteristics under diverse conditions.

## Introduction

The cement sheath is a crucial component in oil and gas well construction, filling the annular space between the borehole and casing. Its primary functions are to seal the wellbore, preventing fluids such as oil, gas, and water from migrating along the outside of the casing, protecting the reservoir and surrounding environment, and providing structural support to the wellbore. During drilling and completion operations, cement slurry is injected into the annular space between the borehole and casing, where it hardens to form a cement sheath. This sheath needs to exhibit good mechanical strength, adhesion, and low permeability to effectively withstand various factors like external pressure changes, temperature fluctuations, and chemical corrosion. Additionally, the cement sheath must maintain stable sealing performance throughout the well's lifespan to prevent any fluid leakage. This stability is vital for protecting the subsurface reservoirs and ensuring long-term well productivity. In petroleum engineering, the permeability of the cement sheath is critically important, as it directly affects the sealing effectiveness of the well. If the cement sheath's permeability increases, fluids may penetrate

**Funding:** The author(s) received no specific funding for this work.

**Competing interests:** On behalf of all authors, disclose any competing interests that could be perceived to bias this work—acknowledging all financial support and any other relevant financial or non-financial competing interests.

through it, potentially leading to fluid migration between the reservoir and surrounding formations. This can result in contamination between formations and could even harm the surrounding ecological environment [1,2]. As the demand for oil and gas well integrity continues to grow domestically and internationally, petroleum companies and researchers worldwide have conducted extensive work on cement sheath pressure evaluation. The U.S. Department of Mineral Resources requires diagnostic testing for any oil or gas well with cement sheath pressure. Based on the test results, it makes decisions regarding pressure release or continued operations [3–5].

Crow et al. [6] examined a well over 30 years old in a natural $CO_2$ reservoir, using downhole and experimental equipment to assess its wellbore sealing performance and analyze its wellbore integrity. Tao et al. [7] investigated the reasonable range of parameters related to the wellbore, applying the SCP model to obtain an approximate range of effective permeability based on recovered annular pressure history data. Kalam et al. [8] conducted numerical simulation studies using finite element methods on wellbore annuli with gas leakage under the influence of dynamic loads. They found that under dynamic loads, the risks of tensile and shear failures increase with the increase of Young's modulus and Poisson's ratio, and lower cement mechanical strength increases the risk of cement shear failure. Rocha-Valadez et al. [9] developed a model to predict sustained casing pressure (SCP) using early pressure recovery data based on the thermodynamics of transport processes and systems. This model demonstrated significant reductions in testing time and limited gas accumulation and risks when used as a prediction tool, enhancing safety and reducing time consumption during testing. Vu et al. [10] proposed a new gas transport model to study the impact of cement components on cement sheath integrity. Langley et al. [11] successfully designed and implemented flexible self-healing cement in approximately 250 wells in the DJ Basin, reducing SCP instances to 2%. Nafikova et al. [12] demonstrated through examples that using flexible self-healing cement systems for casing pressure repair is successful. Mwang et al. [13] designed a novel casing surface for offshore gas wells with microannuli-induced SCP issues, increasing the casing-cement interface (CCI) length under the same cement column length to inhibit fluid flow through CCI.

Research on sustained cement sheath pressure primarily focuses on the theoretical prediction of cement sheath pressure. Various mathematical models are established to simulate and predict the process of cement sheath pressure, and their accuracy is validated by comparing the theoretically calculated pressure change curves with the wellhead pressure change curves obtained from production sites. The results obtained from mathematical models must be compared with on-site production data, highlighting the importance of the rationality and accuracy of these models. Existing mathematical models for casing cement sheath pressure simulation can be categorized into three types: simple models that do not consider gas migration and pipe deformation [14–18]; complex models that consider gas bubble migration but not pipe deformation [19–21]; and models that consider pipe deformation but not gas migration [22–25]. The first model type assumes a constant gas flow velocity from the leak point to the wellhead, directly affecting the model's accuracy. The second type of mathematical model for cement sheath pressure includes two-phase flow models and bubble models, with the former mainly using finite difference methods requiring differentiation of the liquid column segment. During the calculation process, as the wellhead annular pressure continuously changes, the liquid is compressed, resulting in a change in the length of the liquid column segment. This necessitates continuous updates to the differentiation of the fluid column segment, affecting the accuracy of the calculated wellhead annular pressure values. The latter calculates by superimposing individual bubbles, which may not align with actual conditions and could result in an undersized annular volume. The third type of model improves accuracy but leads to an overestimation of the calculated annular volume. Additionally, more research is needed

on key parameters such as formation leakage rate, liquid level height, and comprehensive permeability of cement sheath.

In light of this, this paper establishes a finite element analysis model for cement sheath seepage, thoroughly analyzing the influence of key parameters on leakage for intact cement sheaths, cement sheaths with longitudinal cracks, and cement sheaths with gaps at the casing-cement interface, aiming to enhance the safety and reliability of cement sheaths.

# Models and equations

## Mathematical equations

In the construction of the finite element analysis model for cement sheath percolation, the physical field of Comsol Multiphysics is employed, specifically selecting porous medium flow. This primarily encompasses the physical field interfaces of Darcy's law, the Brinkman equation, free and porous medium flow, and two-phase flow.

(1) Darcy's law

Darcy's law, primarily utilized for addressing slow flow speeds within porous media, is significantly influenced by the resistance of void friction. The cement sheath, characterized as a low-permeability porous medium, necessitates the use of Darcy's law interface. This interface requires the medium to have extremely low permeability and minuscule pores. The governing equation for the physical field interface of Darcy's law is as follows [26–28]:

$$
\begin{cases}
\dfrac{\partial}{\partial t}(\rho \xi) + \nabla \cdot (\rho u) = Q_m, \\[2mm]
u = -\dfrac{\kappa}{\mu} \nabla p, \\[2mm]
\rho = \dfrac{pM}{RT}.
\end{cases}
\tag{1}
$$

Where, $\kappa$ is the permeability of the porous media, $\xi$ is the porosity, $Q_m$ is the quality source item, $R$ is the ideal gas constant.

(2)Brinkman equation

The Brinkman equation serves as a model for delineating fluid flow within a porous medium. It extends from Darcy's law, incorporating the phenomenon of seepage within such a medium. The fundamental formulation of the Brinkman equation is presented as follows [29]:

$$
\mu \left( \frac{\partial^2 u}{\partial x_i \partial x_j} - \frac{1}{\lambda^2} u_i \right) = -\frac{\partial p}{\partial x_i} + \rho g_i - \mu \frac{\partial u_i}{\partial t}.
\tag{2}
$$

Where $u$ is the velocity field vector, $x_i$ is the spatial coordinates, $\mu$ is the dynamic viscosity of the fluid, $\lambda$ is the penetration length of the porous medium, $p$ is the pressure, $\rho$ is the fluid density, $g_i$ is the gravity, $t$ is the time.

## Computational domain

In accordance with Darcy's Law, we incorporated the attributes of pressure distribution, pressure difference, and permeability of the cement sheath seepage to devise a finite element simulation model for intact cement sheath seepage. This model is visually represented in Fig 1, illustrating the mechanical model of the intact cement sheath. Table 1 outlines the primary dimensional parameters and the inlet/outlet conditions of the cement sheath. Specifically, the cement sheath measures 50 meters in length, has an outer diameter of 218.5 mm, and an inner

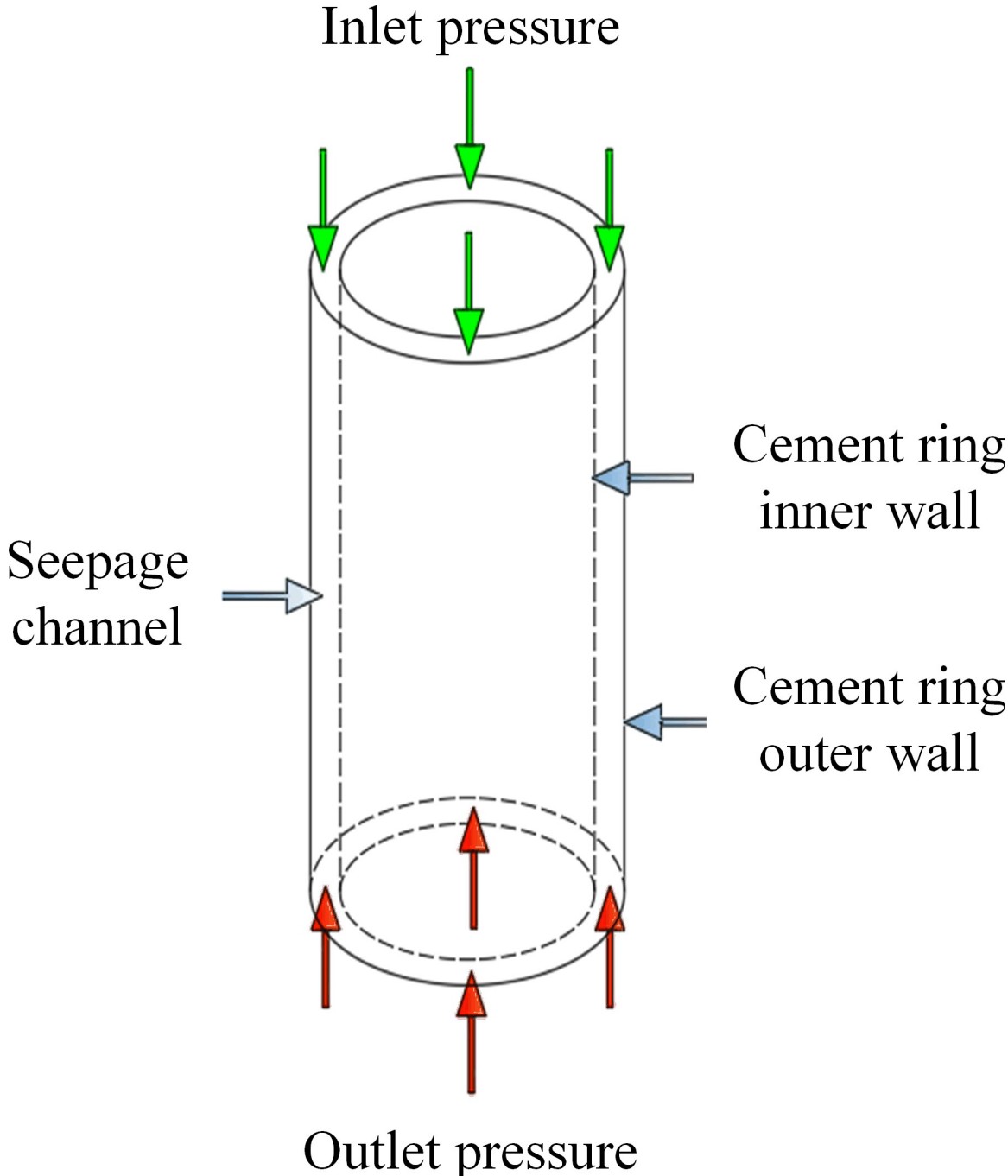

**Fig 1. Schematic representation of the cement sheath computational model.**

diameter of 177.8 mm. The matrix permeability of the cement sheath is 0.01 mD, with an inlet pressure of 120 MPa and an outlet pressure of 100 MPa.

### Grid sensitivity check

As depicted in Fig 2, to guarantee the computational precision of the simulated seepage flow, the grid was initially examined, followed by a two-fold encryption of the entire well section's

**Table 1. Cement sheath parameter.**

| Parameters/units | Value |
|---|---|
| Length/m | 50 |
| External diameter/mm | 218.5 |
| Internal diameter /mm | 177.8 |
| Permeability/mD | 0.01 |
| Inlet pressure/MPa | 120 |
| Outlet pressure /MPa | 100 |

grid. This approach constrained the maximum unit to 0.001 m, the maximum unit growth rate to 1.1, the minimum cell to $1 \times 10^{-4}$ m, and established the curvature factor at 0.2. Each group's computational duration was set at 0.5 hours.

## Model verification

Based on the cement sheath model illustrated in Fig 1, this study has developed two additional models: a cement sheath model with a longitudinal crack, and a cement sheath model with a gap at the casing joint surface. The validation process for these three models is outlined as follows.

(1) Complete cement sheath model validation

Fig 3 illustrates the distribution diagrams of the complete cement sheath seepage pressure field and the Darcy velocity field. The seepage pressure within the complete cement sheath uniformly decreases from the inlet to the outlet, ranging from a maximum of 120 MPa at the inlet to a minimum of 100 MPa at the outlet, aligning with the parameters set in the model. Additionally, the Darcy velocity field across the complete cement sheath remains consistent from the inlet to the outlet, at approximately $3.91 \times 10^{-9}$ m/s. This uniformity suggests that the seepage velocity is consistent throughout the cement sheath, unaffected by other variables.

(2)The cement sheath finite element modeling in the presence of longitudinal cracks

The cement sheath percolation mechanics model has been expanded to incorporate a cleft structure, resulting in a modified model that accounts for a longitudinal fissure. A simulated mechanical model of cement sheath seepage, factoring in longitudinal cracks, is depicted in Fig 4.

The parameters of the study include a fissure direction parallel to the axial cement sheath, a cement sheath length of 1m, and a cement sheath outer and inner diameter of 218.5 mm and 177.8 mm respectively. The cement sheath matrix permeability is set at 0.01 mD, with a fissure length of 0. The finite element simulation results are depicted in Fig 5, using a cement sheath of 1 mm, upper pressure of 120 MPa, and a lower pressure of 100 MPa.

The figure illustrates a gradual decrease in the surface pressure of the cement sheath from the inlet to the outlet, with values of 120 MPa and 100 MPa, respectively, which align with the model's parameters. A comparison of the pressure distribution contour map without longitudinal cracks reveals that the presence of longitudinal cracks results in increased flow velocity at the crack, thereby reducing the pressure change gradient in the affected area. Furthermore, the Darcy velocity field of the cement sheath exhibits significant alterations at both ends of the crack, with the Darcy velocity at the crack substantially exceeding the flow velocity of the cement sheath's porous medium in the absence of cracks, corroborating theoretical expectations.

(3)The cement sheath finite element modeling in the presence of longitudinal cracks

Based on the comprehensive cement sheath percolation mechanics model, this study incorporates the casing model to establish the annular structure between the casing and the cement

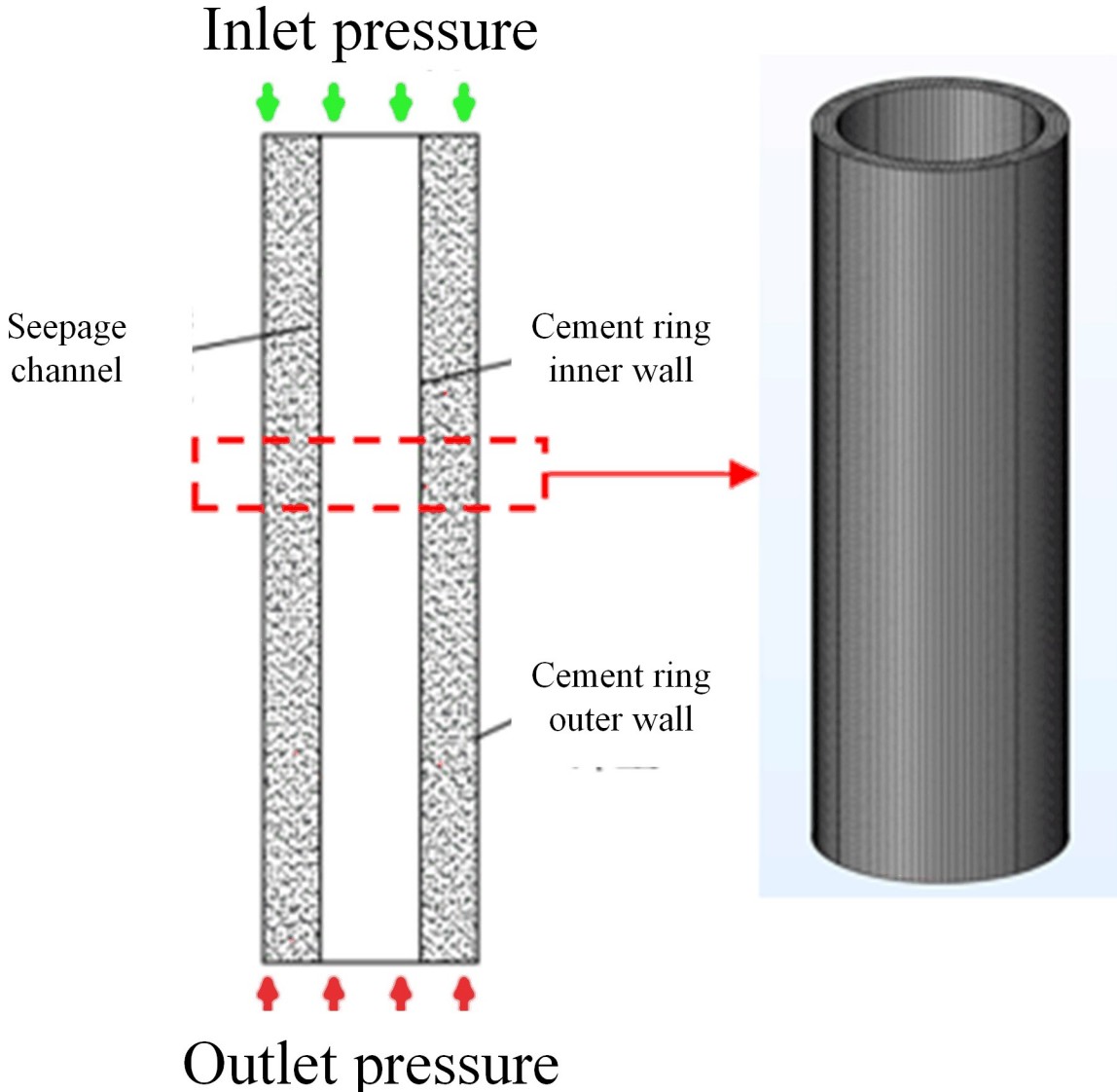

Inlet pressure

Seepage channel

Cement ring inner wall

Cement ring outer wall

Outlet pressure

**Fig 2. Grid sensitivity check.**

sheath. Consequently, a refined cement sheath percolation mechanics model that includes the casing joint surface is developed, as illustrated in Fig 6.

The parameters depicted in Fig 7 include a cement sheath with a length of 1 m, an outer diameter of 218.5 mm, and an inner diameter of 177.8 mm. The matrix permeability of the cement sheath is 0.01 mD. Additionally, the axial length is 0.8 m, the gap thickness is 0.8 mm, and the pressures at the upper and lower ends of the cement sheath are 120 MPa and 100 MPa, respectively.

The figure illustrates a decrease in the surface pressure of the cement sheath from the inlet to the outlet, commencing at an inlet pressure of 120 MPa and culminating at an outlet pressure of 100 MPa, a pattern that aligns with the model settings. When compared to the intact cement sheath model, it becomes evident that the gaps present between the cement sheath and the casing-cement interface result in a significantly smaller rate of pressure change in the central segment compared to the segment devoid of cracks. This leads to minimal alterations in

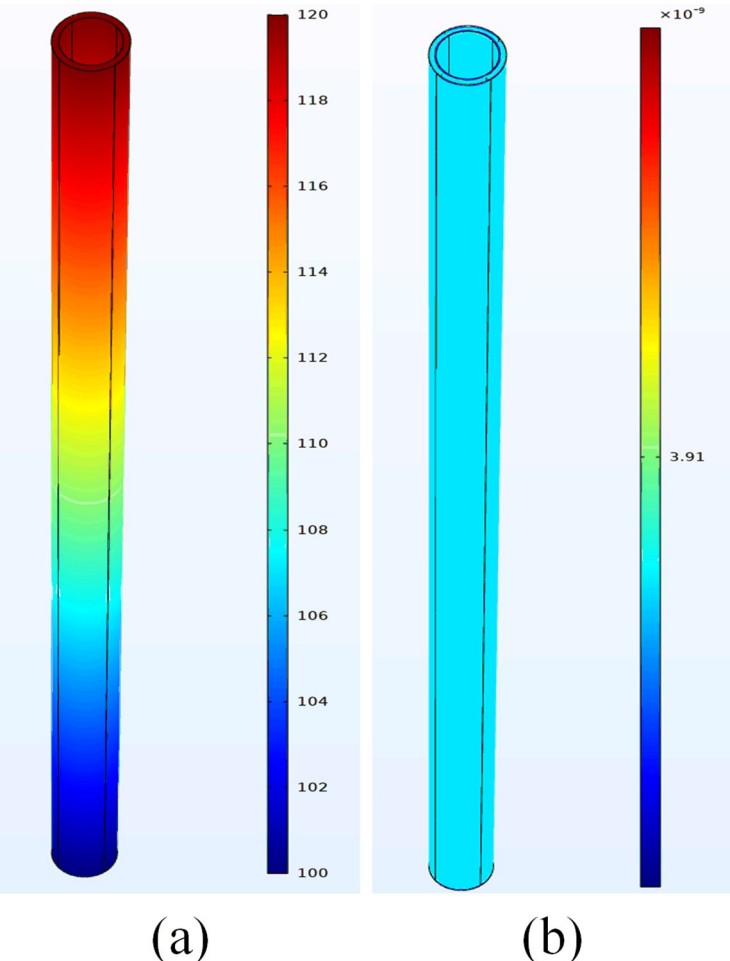

**Fig 3.** Complete cement sheath seepage pressure field and Darcy velocity field diagram: (a) pressure distribution; (b) Darcy velocity distribution.

the pressure gradient and an accelerated increase in flow velocity at the interface. However, there are conspicuous discontinuities in both pressure and Darcy velocity at the extremities of the cement sheath and the casing-cement interface. These discontinuities can be attributed to the gaps between the cement sheath and the casing-cement interface, which cause disruptions in both pressure and velocity gradients.

## Results and discussion

This paper addresses the issue of fluid seepage within the cement sheath downhole. The characteristics of this seepage are examined in depth, taking into account factors such as the inherent properties of the cement sheath, the state of internal fissures, the condition of gaps, and the characteristics of the fluid. A simulation model of cement sheath seepage is established using the finite element method, enabling a comprehensive analysis of the influences of variables such as the length and permeability of the cement sheath, the structure of the gap, pressure differences, and fluid characteristics.

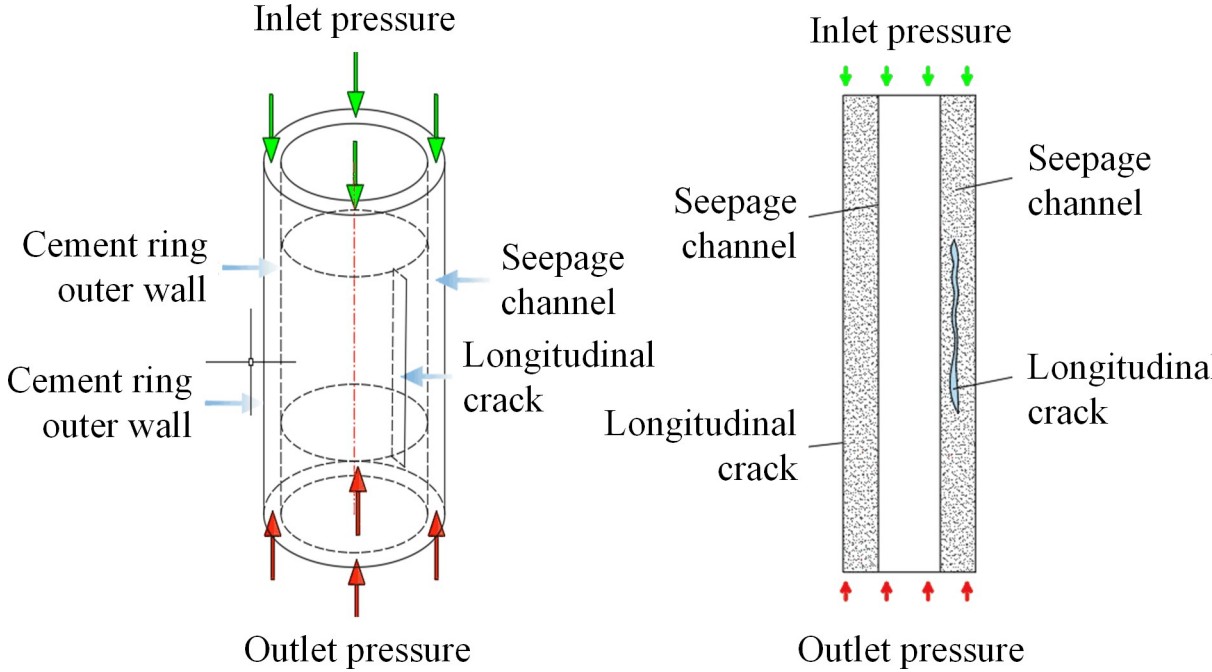

**Fig 4. A cement sheath seepage simulation model considering longitudinal cracks.**

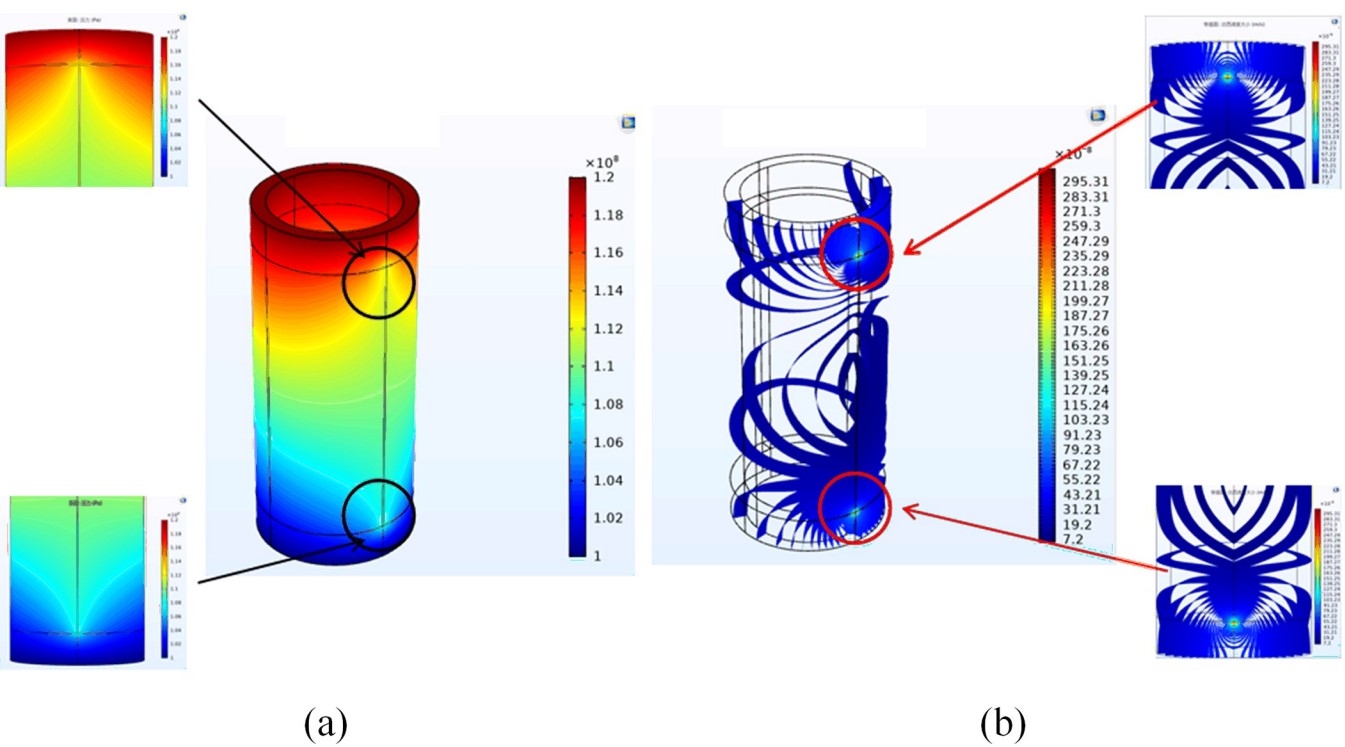

(a)                                                            (b)

**Fig 5.** Considers the cement sheath seepage pressure field and Darcy velocity field diagram: (a) pressure distribution; (b) Darcy velocity distribution.

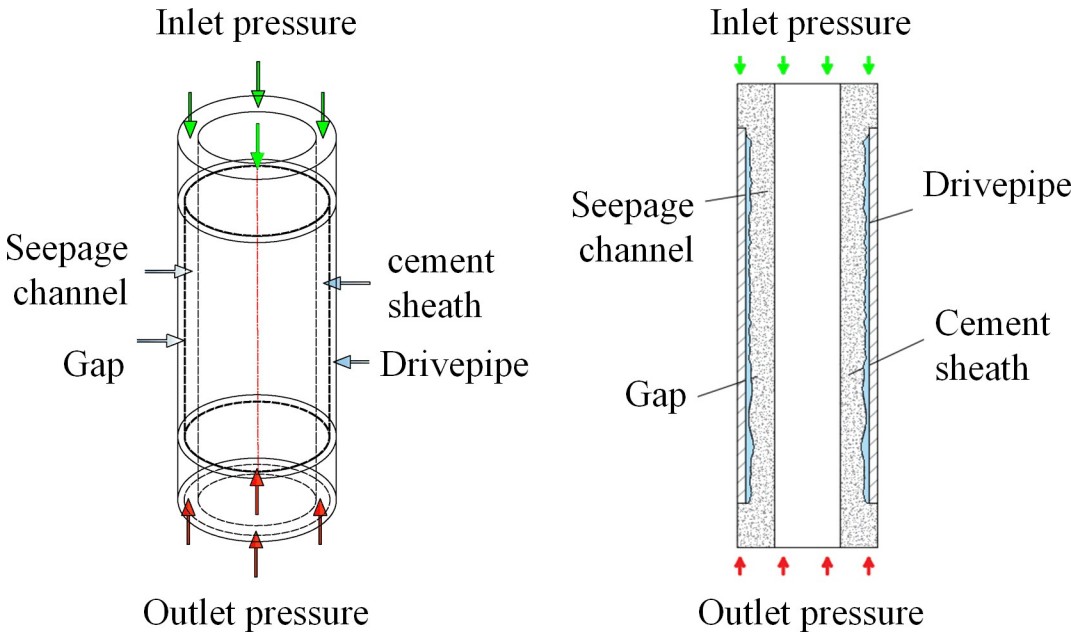

**Fig 6. Simulation model of cement sheath seepage between sleeve tubes.**

## Analysis of the key parameters affecting the complete cement sheath seepage flow rate

The seepage process is a nonlinear hydrodynamic problem for cement sheath seepage under different permeability and pressure difference conditions. The key seepage parameters of cement sheath for two different structural parameters, and the geometric parameters of cement sheath are shown in Table 2 [30].

(1) Analysis of key parameters when using liquid as seepage medium

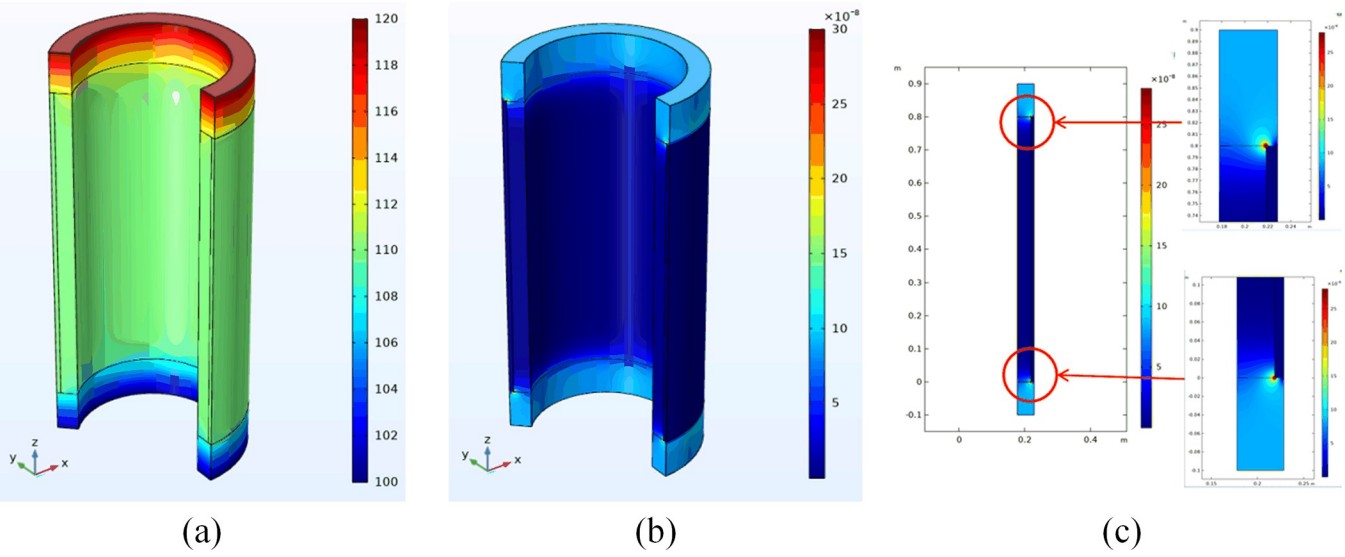

**Fig 7.** Cement sheath seepage pressure field and Darcy velocity field map on the casing bonding surface: (a) pressure distribution; (b) Darcy velocity distribution; (c) Darcy velocity distribution of cement sheath section liquid seepage.

**Table 2. The cement sheath structures.**

|  | Internal diameter (mm) | External diameter (mm) | Length(m) | Penetrance (mD) | Inlet pressure(MPa) | Outlet pressure (MPa) |
|---|---|---|---|---|---|---|
| cement sheath-1 | 177.8 | 218.5 | 50m | 0.01 | 120 | 120 |
| cement sheath-2 | 244.5 | 313.58 | 50m | 0.01 | 100 | 100 |

Fig 8A–8C provides a detailed illustration of how three crucial parameters—matrix permeability, pressure difference, and cement sheath length influence the cement sheath seepage flow. The analysis of these diagrams supports the following conclusions:

Fig 8A illustrates that as permeability linearly increases from 0 to 10 millidarcies (mD), both Cement Sheath 1 and Cement Sheath 2 demonstrate a linear increase in Darcy seepage

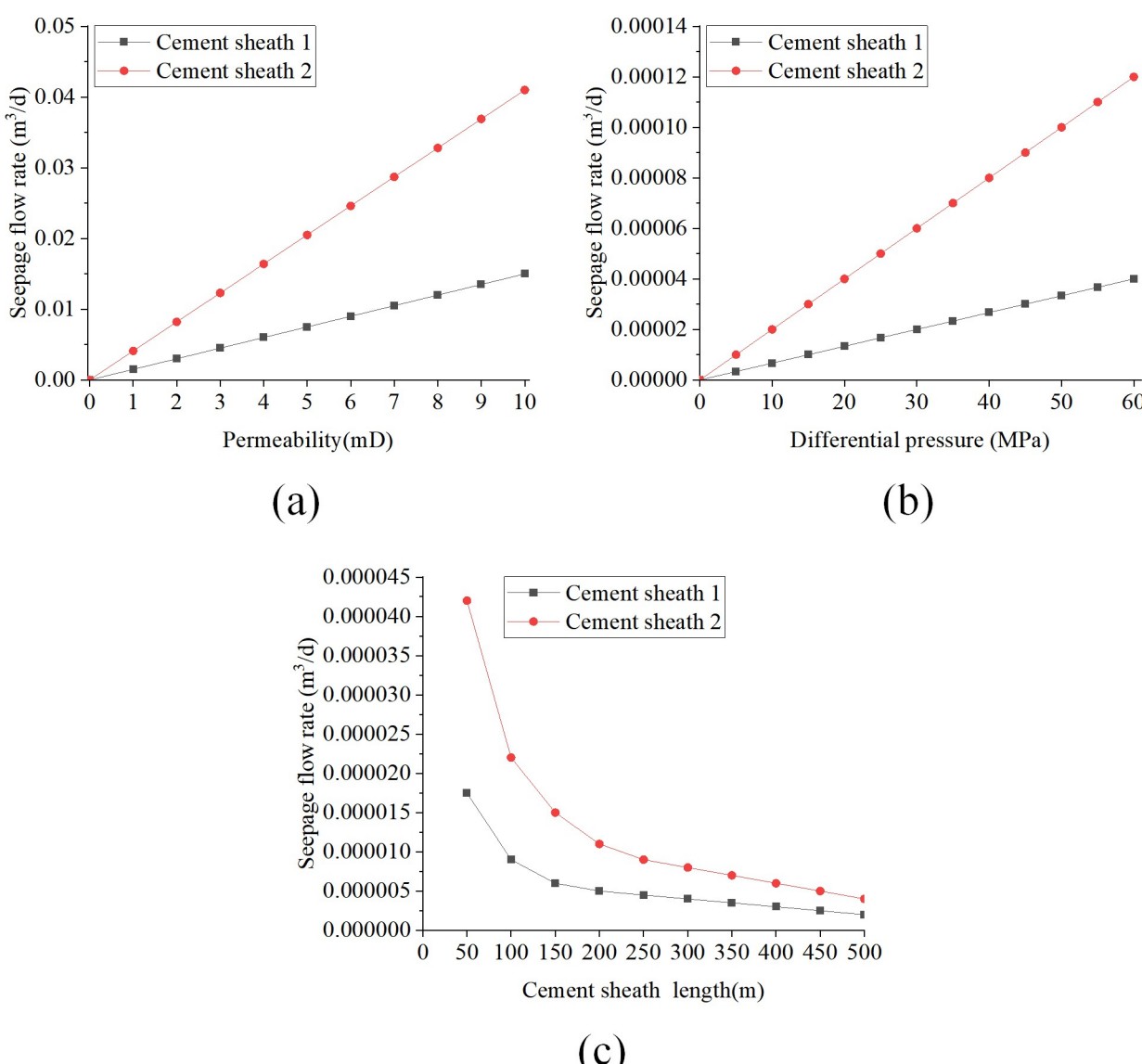

**Fig 8.** Effect of key parameters on seepage flow when liquid is used as seepage medium: (a) change of seepage flow with permeability; (b) change of seepage flow with pressure difference; (c) change of seepage flow with length.

flow. Specifically, the seepage flow in Cement Sheath 1 rises from 0 to $1.54\times10^{-2}$ m$^3$/d, and in Cement Sheath 2 from 0 to $3.69\times10^{-2}$ m$^3$/d. This trend suggests that higher permeability enhances fluid flow within the cement sheaths, thereby increasing the seepage flow rate. Fig 8B shows that increasing the pressure difference from 0 to 60 megapascals (MPa) results in a similar linear growth in the Darcy seepage flow for both sheaths. Upon reaching 60 MPa, the seepage flow in Cement Sheath 1 increases from 0 to $5.14\times10^{-5}$ m$^3$/d, and in Cement Sheath 2 to $1.23\times10^{-4}$ m$^3$/d. This indicates that a higher pressure difference acts as a driving force for fluid flow within the cement sheaths, which in turn increases the seepage flow rate. Conversely, as depicted in Fig 8C, increasing the length of the cement sheath from 50 m to 500 m is associated with a nonlinear decrease in Darcy seepage flow for both Cement Sheath 1 and Cement Sheath 2. At a length of 500 m, the seepage flow in Cement Sheath 1 declines to $1.71\times10^{-6}$ m$^3$/d, and in Cement Sheath 2 to $4.09\times10^{-5}$ m$^3$/d. This pattern indicates that a longer cement sheath may impede fluid flow, thereby reducing the seepage flow rate.

(2) Analysis of key parameters with gas as seepage medium

Fig 9A–9C demonstrate the influence of cement sheath matrix permeability, pressure differential, and cement sheath length on the seepage flow when gas is the seeping medium. These figures provide a visual representation of how these variables affect the seepage characteristics of the cement sheath. Fig 9A reveals that as the permeability incrementally increases from 0 mD to 10 mD, both cement sheath-1 and cement sheath-2 display a corresponding linear increase in Darcy seepage flow. Notably, when the permeability reaches 10 mD, the seepage flow for cement sheath-1 and cement sheath-2 escalates to 0.94 m$^3$/d and 2.49 m$^3$/d, respectively. In Fig 9B, with the pressure differential linearly escalating from 0 MPa to 60 MPa, both cement sheath-1 and cement sheath-2 exhibit a nonlinear increase in Darcy seepage flow. Moreover, the seepage flow rate progressively accelerates as the pressure differential intensifies. At a pressure differential of 60 MPa, the seepage flow for cement sheath-1 and cement sheath-2 amounts to $4.29\times10^{-3}$ m$^3$/d and $1.02\times10^{-2}$ m$^3$/d, respectively. Fig 9C demonstrates that as the cement sheath length linearly expands from 50 m to 500 m, both cement sheath-1 and cement sheath-2 display a nonlinear reduction in Darcy seepage flow. When the cement sheath length extends to 500 m, the seepage flow for cement sheath-1 and cement sheath-2 diminishes to $1.05\times10^{-4}$ m$^3$/d and $2.51\times10^{-4}$ m$^3$/d, respectively.

## Analysis of cement sheath seepage in the presence of longitudinal cracks

To ascertain the impact of internal fissures in the cement sheath on seepage flow, a simulation of seepage under conditions of a banded crack in the cement sheath is conducted.

(1) Analysis of key parameters when using liquid as seepage medium

Fig 10A–10C illustrate the effects of varying parameters such as permeability, Darcy seepage flow rate, and other characteristics of a cement sheath matrix with longitudinal fractures when using a liquid seepage medium under different conditions of fracture width, pressure differential, and cement sheath length. These images provide a visual representation of the influence of fracture width, cement sheath length, and fracture permeability on the seepage flow rate through the cement sheath. Fig 10A demonstrates that increasing the fracture width from 1 mm to 10 mm results in a nonlinear increase in the Darcy seepage flow rate. For each millimeter increase in fracture width, the seepage flow rate rises; however, this rate of increase diminishes as the fracture width expands. At a fracture width of 10 mm, the seepage flow rate peaks at $1.36\times10^{-3}$ m$^3$/d. According to Fig 10B, the Darcy seepage flow rate decreases gradually as the cement sheath length increases linearly from 1 m to 5 m. The rate of decline in the seepage flow rate diminishes over this range, decreasing from $1.18\times10^{-3}$ m$^3$/d to $5.3\times10^{-4}$ m$^3$/d as the length extends. Fig 10C indicates that an increase in fracture permeability from 0 mD to 10

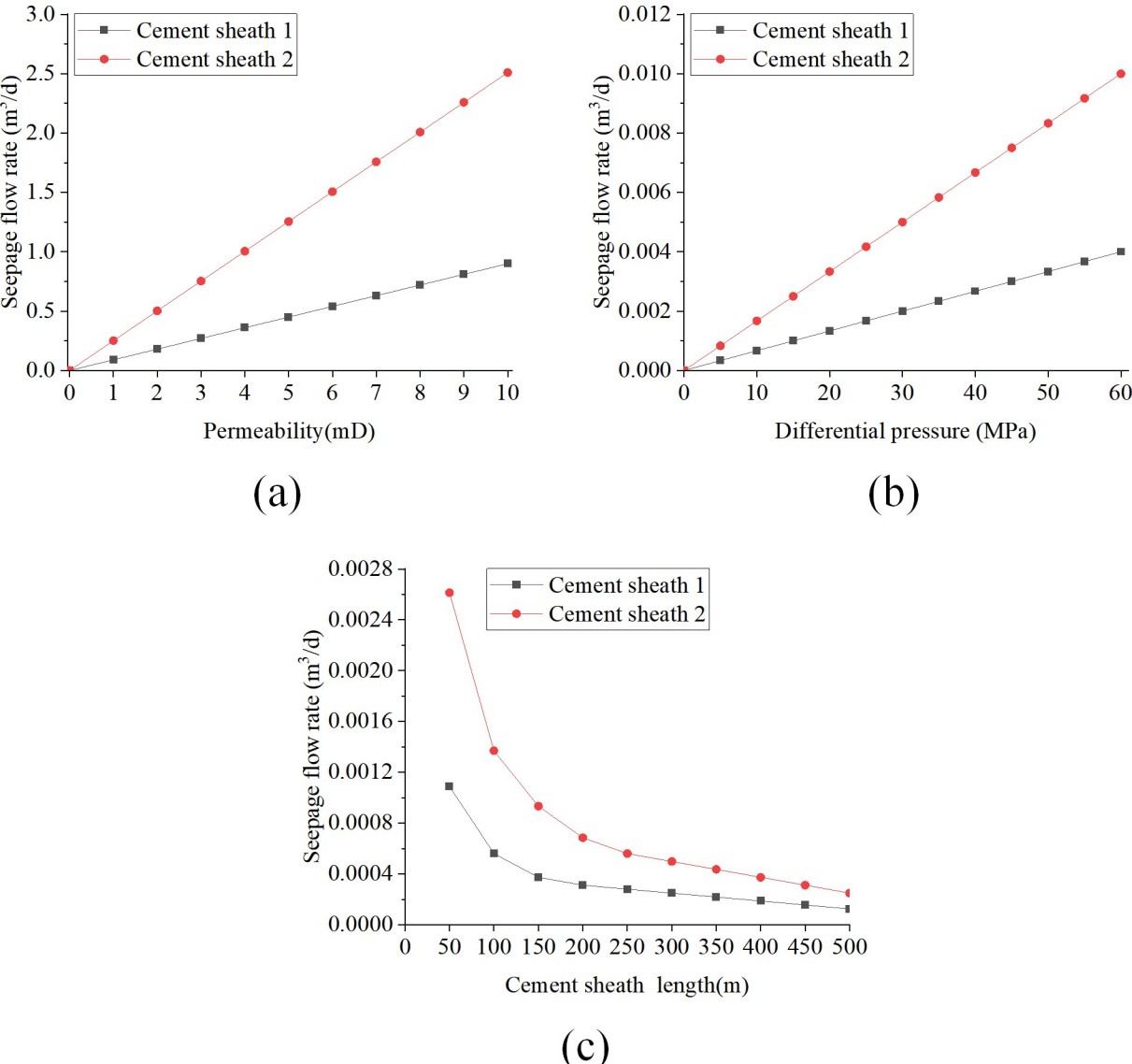

**Fig 9.** Effect of the key parameters on the seepage flow rate when the gas is used as the seepage medium: (a) plot of seepage flow with permeability; (b) plot of seepage flow with pressure difference; (c) plot of seepage flow with length.

mD leads to a rising trend in the Darcy seepage flow rate. While the rate continues to increase, the acceleration of the increase slows as the permeability approaches 10 mD, culminating in a seepage flow rate of $1.28 \times 10^{-3}$ m³/d.

(2) Analysis of key parameters with gas as seepage medium

Fig 11A–11C illustrate the impact of various parameters such as permeability, pressure differential, and cement sheath length on the seepage flow rate of a cement sheath matrix containing longitudinal fractures when gas is used as the seepage medium. Fig 11A demonstrates that as fracture width increases linearly from 1 mm to 10 mm, the Darcy seepage flow rate exhibits a nonlinear growth trend. The data show a rapid initial increase in flow rate as fracture width expands, followed by a deceleration in the rate of increase. At a fracture width of 10 mm, the flow rate reaches $6.9 \times 10^{-2}$ m³/d, indicating a substantial rise. In Fig 11B, the variation in Darcy seepage flow rate is analyzed as the cement sheath length increases from 1 m to 5 m.

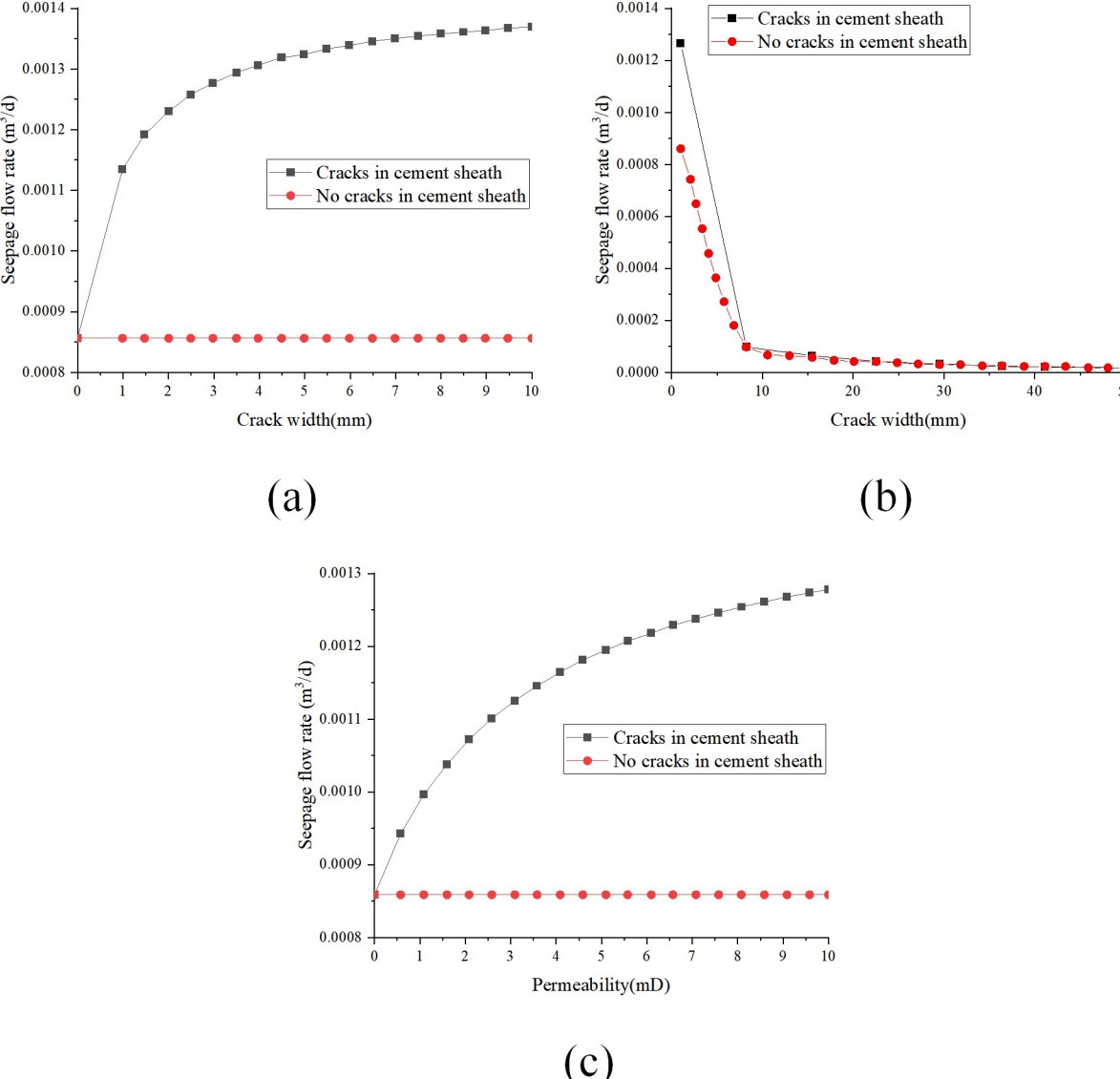

**Fig 10.** The influence of key parameters of cement sheath on seepage flow in the presence of longitudinal fissure and fluid as the seepage medium: (a) change of seepage flow with fissure width; (b) change of seepage flow with cement sheath length; (c) change of seepage flow with fissure permeability.

The flow rate decreases progressively with increasing cement sheath length, exhibiting a linear decay trend. Specifically, the seepage flow rate diminishes from $5.9 \times 10^{-2}$ m³/d at 1 m to $2.5 \times 10^{-2}$ m³/d at 5 m. Fig 11C presents the changes in Darcy seepage flow rate with a linear increase in fracture permeability from 0 mD to 10 mD. The seepage flow rate increases with rising fracture permeability, yet the acceleration rate of increase slows over time. At a permeability of 10 mD, the flow rate peaks at $6.4 \times 10^{-2}$ m³/d, showcasing a significant enhancement in seepage.

### Analysis of cement sheath seepage with the with the gap in the casing joint surfacee

(1)Analysis of key parameters when using liquid as seepage medium

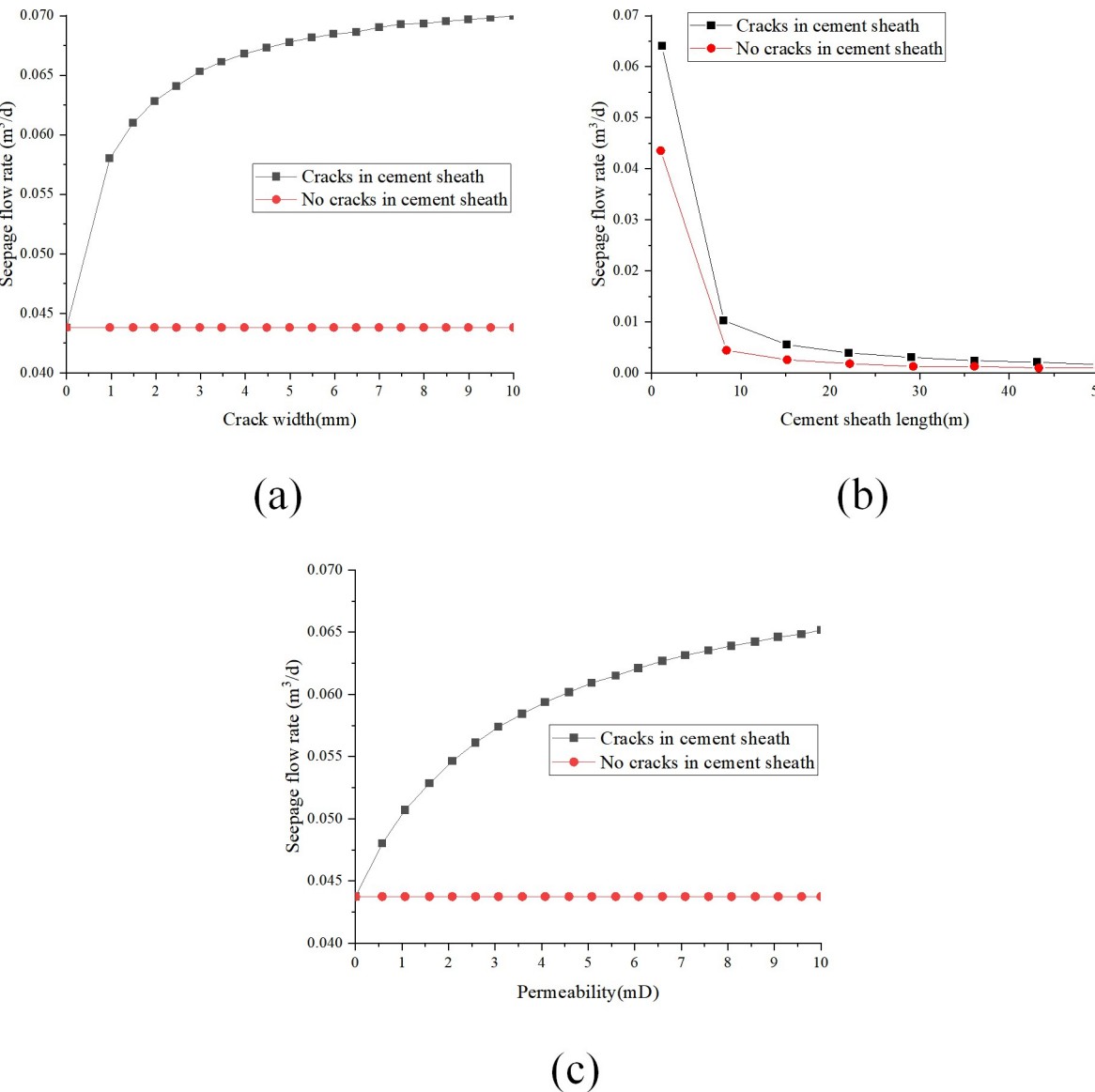

**Fig 11.** The influence of cement sheath key parameters on the seepage flow in the presence of longitudinal fissure and gas as the seepage medium: (a) change of seepage flow with fissure width; (b) change of seepage flow with cement sheath length; (c) change of seepage flow with fissure permeability.

Fig 12A–12C illustrate the influence of parameters such as permeability, pressure differential, and cement sheath length on the seepage flow rate of the cement sheath matrix with gaps at the casing-cement interface, using liquid as the seepage medium.

Fig 12A reveals that as the fracture width linearly increases from 1 mm to 10 mm, the Darcy seepage flow rate follows a nonlinear growth pattern. Specifically, the augmentation in fracture width results in a swift increase in seepage flow rate, although the rate of increase progressively diminishes. When the fracture width attains 10 mm, the seepage flow rate reaches $3.6\times10^{-3}$ $m^3$/d, underscoring the substantial effect of fracture width enlargement on the enhancement of seepage flow rate. Fig 12B shows that the Darcy seepage flow rate gradually declines as the cement sheath length linearly increases from 1 m to 5 m. Importantly, the rate of decrease in

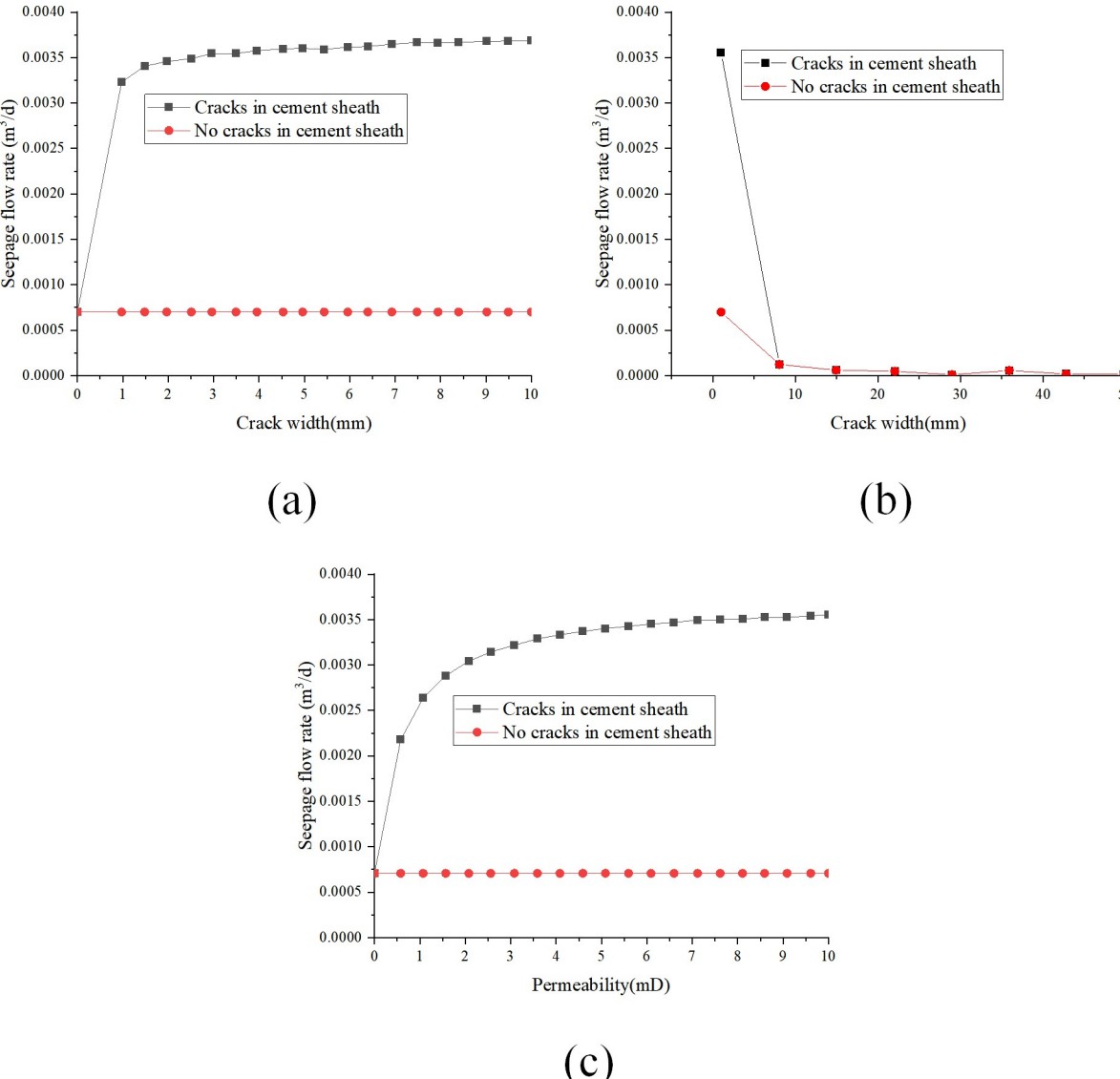

**Fig 12.** The influence of cement sheath key parameters with gaps in the adhesive surface of the sleeve on the seepage flow with the liquid is used as the seepage: (a) change of seepage flow with crack width; (b) change of seepage flow with cement sheath length; (c) change of seepage flow with fissure permeability.

seepage flow rate progressively decelerates as the cement sheath length increases. An increase in cement sheath length from 1 m to 5 m results in a decrease in seepage flow rate from $3.5 \times 10^{-3}$ m³/d to $2.78 \times 10^{-3}$ m³/d, indicating that alterations in cement sheath length exert a certain influence on seepage flow rate. As depicted in Fig 12C, the Darcy seepage flow rate generally escalates with a linear increase in fracture permeability from 0 mD to 10 mD. Nevertheless, the rate of increase in seepage flow rate gradually slows with the continuous increase in permeability. Upon reaching a fracture permeability of 10 mD, the seepage flow rate achieves $3.3 \times 10^{-3}$ m³/d, suggesting that fracture permeability has a regulatory impact on seepage flow rate to some degree.

(2)Analysis of key parameters with gas as seepage medium

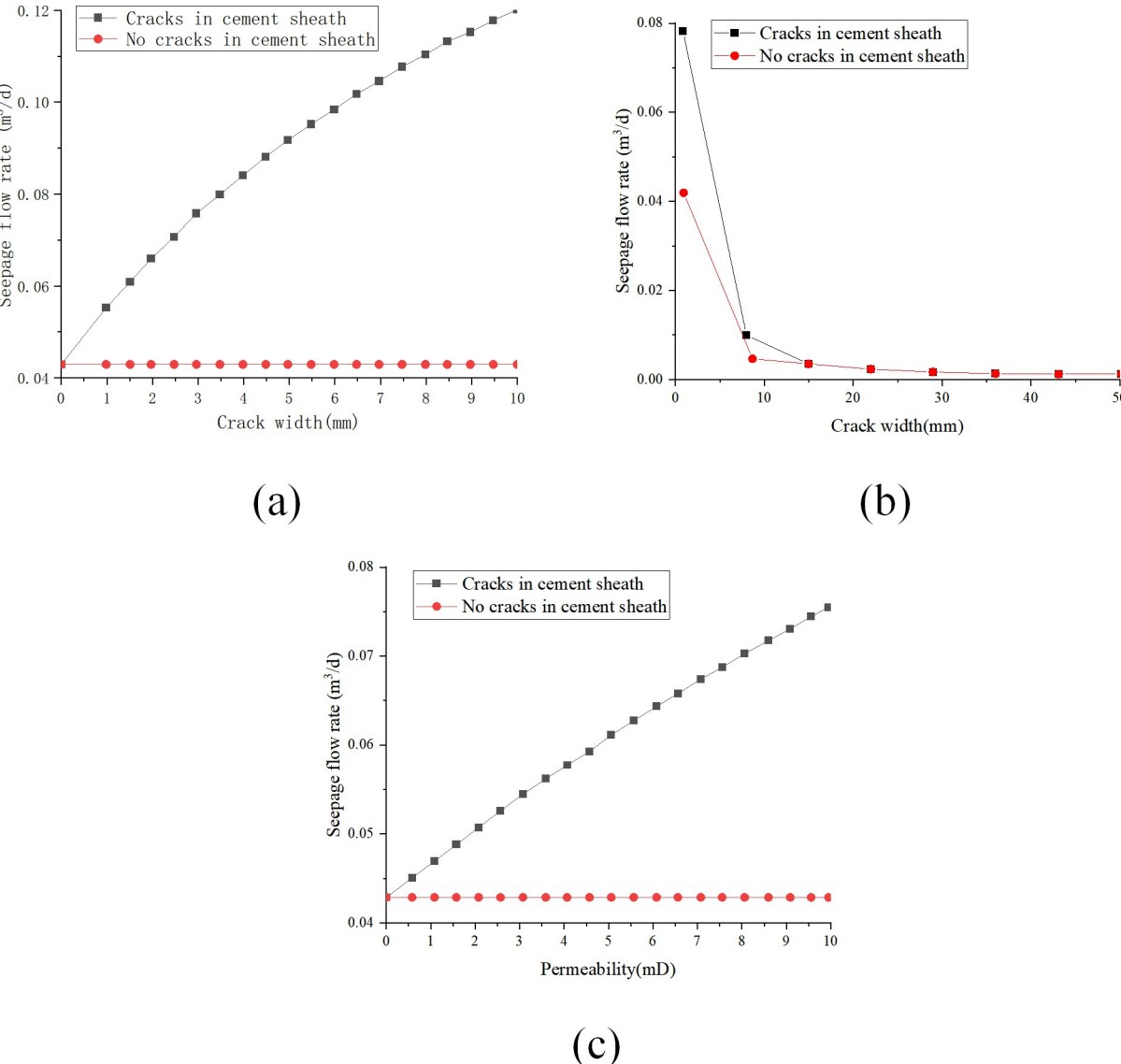

**Fig 13.** The influence of cement sheath key parameters with gaps in the adhesive surface of the sleeve on the seepage flow with the gas is used as the seepage: (a) change of seepage flow with crack width; (b) change of seepage flow with cement sheath length; (c) change of seepage flow with fissure permeability.

Fig 13A–13C illustrate the impact of parameters such as permeability, pressure differential, and cement sheath length on the seepage flow rate of the cement sheath matrix when gas is utilized as the seepage medium. These figures underscore the substantial influence of parameter variations on the seepage flow rate.

Fig 13A reveals a nonlinear increase in the Darcy seepage flow rate with the expansion of fracture width from 1 mm to 10 mm. Specifically, the enlargement of fracture width results in an increase in seepage flow rate from $0.42 \times 10^{-1}$ m³/d of 1 mm to $1.19 \times 10^{-1}$ m³/d, with the rate of increase declining as the fracture width broadens. This finding suggests that while an increase in fracture width positively affects the seepage flow rate, the rate of enhancement decreases over time. Fig 13B displays a diminishing trend in the Darcy seepage flow rate with a linear expansion in cement sheath length from 1 m to 5 m. This observation implies that an

increase in cement sheath length negatively affects the seepage flow rate. However, as the length extends, the rate of decrease in seepage flow rate progressively decelerates, suggesting that although the augmentation of cement sheath length reduces the seepage flow rate, this effect diminishes over time. Fig 13C shows an ascending trend in Darcy seepage flow rate as fracture permeability increases from 0 mD to 10 mD. However, with a further increase in permeability, the rate of increase in seepage flow rate gradually decelerates. This finding suggests that while increasing fracture permeability can enhance the seepage flow rate to a certain extent, this effect gradually diminishes as permeability continues to increase.

## Conclusion

This study employed finite element methods to develop a simulation model for cement sheath seepage and analyzed key parameters influencing seepage flow, such as cement sheath length, permeability, gap structure, pressure difference, and fluid characteristics. The primary conclusions are as follows:

Regarding liquid seepage, an increase in permeability and pressure difference enhances the flow rate, whereas an increase in length reduces it. In the case of gas seepage, an increase in permeability and pressure difference also increases the flow rate, but an increase in length leads to a nonlinear decrease. These findings contribute to the understanding and prediction of seepage characteristics of cement sheaths under various conditions.

In a liquid medium, factors such as fracture width, cement sheath length, and fracture permeability influence the seepage flow rate. Increasing fracture width results in nonlinear growth of the flow rate with diminishing acceleration. Increasing cement sheath length causes a gradual decrease in seepage flow rate, with the rate of decline slowing down. Increasing fracture permeability generally raises the seepage flow rate, but the rate of increase decelerates over time. In a gas medium, the influencing factors and trends are consistent with those in a liquid medium.

The study examined cement sheath seepage with gaps at the casing-cement interface. In a liquid medium, increasing fracture width, cement sheath length, and fracture permeability lead to nonlinear growth in Darcy seepage flow rate with decreasing acceleration. In a gas medium, increasing fracture width and permeability result in nonlinear growth in Darcy seepage flow rate with decreasing acceleration. Increasing cement sheath length negatively impacts seepage flow rate; however, as the length increases, the rate of flow rate decline decreases.

## Author Contributions

**Data curation:** Weidong Zhang.

**Formal analysis:** Kewei Xu.

**Investigation:** Jingwei Yang.

**Methodology:** Yangyang Liu.

**Resources:** Wei Xiao.

**Software:** Mingji Wei.

**Validation:** Liqin Qian.

**Writing – original draft:** Luo Wei.

**Writing – review & editing:** Chengyu Xia.

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
