## [Decision Letter · Decision Letter 0]

7 Oct 2024

PONE-D-24-39005Analysis of key parameters influencing the permeability of cement sheath based on multiphysical fieldsPLOS ONE

Dear Dr. xia,

Thank you for submitting your manuscript to PLOS ONE. After careful consideration, we feel that it has merit but does not fully meet PLOS ONE’s publication criteria as it currently stands. Therefore, we invite you to submit a revised version of the manuscript that addresses the points raised during the review process.

We look forward to receiving your revised manuscript.

Kind regards,

Ajaya Bhattarai

Academic Editor

PLOS ONE

Journal requirements: When submitting your revision, we need you to address these additional requirements. 1. Please ensure that your manuscript meets PLOS ONE's style requirements, including those for file naming. The PLOS ONE style templates can be found at https://journals.plos.org/plosone/s/file?id=wjVg/PLOSOne_formatting_sample_main_body.pdf and https://journals.plos.org/plosone/s/file?id=ba62/PLOSOne_formatting_sample_title_authors_affiliations.pdf 2. Please note that PLOS ONE has specific guidelines on code sharing for submissions in which author-generated code underpins the findings in the manuscript. In these cases, we expect all author-generated code to be made available without restrictions upon publication of the work. Please review our guidelines at https://journals.plos.org/plosone/s/materials-and-software-sharing#loc-sharing-code and ensure that your code is shared in a way that follows best practice and facilitates reproducibility and reuse. 3. We note that the grant information you provided in the ‘Funding Information’ and ‘Financial Disclosure’ sections do not match.  When you resubmit, please ensure that you provide the correct grant numbers for the awards you received for your study in the ‘Funding Information’ section. 4. Thank you for stating the following in the Acknowledgments Section of your manuscript: [This research is supported by the Open Foundation of Cooperative Innovation Center of Unconventional Oil and Gas, Yangtze University (Ministry of Education & Hubei Province), No. UOG2024-23]We note that you have provided funding information that is not currently declared in your Funding Statement. However, funding information should not appear in the Acknowledgments section or other areas of your manuscript. We will only publish funding information present in the Funding Statement section of the online submission form. Please remove any funding-related text from the manuscript and let us know how you would like to update your Funding Statement. Currently, your Funding Statement reads as follows:  [The author(s) received no specific funding for this work.] Please include your amended statements within your cover letter; we will change the online submission form on your behalf.

Additional Editor Comments:

The academic editor would like to see the revised manuscript.

Reviewers' comments:

Reviewer's Responses to Questions

**Comments to the Author**

1. Is the manuscript technically sound, and do the data support the conclusions?

Reviewer #1: Yes

Reviewer #2: Yes

2. Has the statistical analysis been performed appropriately and rigorously? 

Reviewer #1: I Don't Know

Reviewer #2: N/A

3. Have the authors made all data underlying the findings in their manuscript fully available?

Reviewer #1: Yes

Reviewer #2: Yes

4. Is the manuscript presented in an intelligible fashion and written in standard English?

Reviewer #1: Yes

Reviewer #2: Yes

5. Review Comments to the Author

Reviewer #1: Report

Title: Analysis of key parameters influencing the permeability of cement sheath based on multiphysical fields

This paper investigates the seepage characteristics of cement sheaths, considering the flow properties of porous media, by utilising finite element analysis in Comsol Multiphysics software. The model described in the present paper seems to have some novelty in the area of fluid flow through porous media (cement sheaths), which may be applicable in diagnostic testing for any oil or gas well with cement sheath pressure. The manuscript is written well and seems suitable for publication in the esteemed journal, but it requires further minor revision to enhance its overall quality.

The following comments must be addressed before making a decision for this manuscript:

1. After the precise review of the paper, it is found that the basic introduction of cement sheath and other keywords are missing. Please incorporate it in the introduction section.

2. Improve the typos errors in whole manuscript. For now:

“CO2” in line 75; “..” in line 182; “(3.9 \\times 10{-9})” in line 190; Check line 206;

“1.54×10-2” in line 279; “from 1 mm to 1.19×10-1 m3/d” in line 417.

3. There are many abbreviations used in this paper. It is advised to include a table for all abbreviations used in the paper.

4. It is also advised to provide citations for all governing equations and utilised formulas. Also, use proper syntax (comma or full stop) at the end of displayed equations

5. Give the references for the chosen numerical range of parameter values of emerging parameters.

6. Try to exclude access use of indent in each paragraph.

7. For instance, address the following articles in context of flow through porous media:

a) https://doi.org/10.1016/j.chaos.2024.114961

b) https://doi.org/10.1002/zamm.202200047

c) https://doi.org/10.1016/j.chaos.2024.114726

d) https://doi.org/10.1016/j.cjph.2024.01.017

b) https://doi.org/10.1016/j.cjph.2024.03.046

Reviewer #2: This reviewer carefully read the manuscript and found it to be well-written. The study focused on the finite element methods to develop a simulation model for cement sheath seepage. It analyzes parameters influencing seepage flow, cement sheath length, permeability, gap structure, pressure difference, and fluid characteristics. However, before accepting the manuscript, the corresponding author/s should make some minor corrections pointed out in the manuscript, including typo errors.

Also, in some parts of the Results and Discussion section, sufficient discussion with recently published citations is not done; only results are described.

Could be accept after minor corrections.

6. PLOS authors have the option to publish the peer review history of their article (what does this mean?). If published, this will include your full peer review and any attached files.

Reviewer #1: No

Reviewer #2: **Yes: **Prof. Dr. Jagadeesh Bhattarai

---

## [Author Response · Author response to Decision Letter 0]

13 Nov 2024

Reviewer #1: Report

Title: Analysis of key parameters influencing the permeability of cement sheath based on multiphysical fields

This paper investigates the seepage characteristics of cement sheaths, considering the flow properties of porous media, by utilising finite element analysis in Comsol Multiphysics software. The model described in the present paper seems to have some novelty in the area of fluid flow through porous media (cement sheaths), which may be applicable in diagnostic testing for any oil or gas well with cement sheath pressure. The manuscript is written well and seems suitable for publication in the esteemed journal, but it requires further minor revision to enhance its overall quality.

The following comments must be addressed before making a decision for this manuscript:

1. After the precise review of the paper, it is found that the basic introduction of cement sheath and other keywords are missing. Please incorporate it in the introduction section.

Dear reviewer

We has been added the basic introduction of cement sheath and other keywords.

The cement sheath is a crucial component in oil and gas well construction, filling the annular space between the borehole and casing. Its primary functions are to seal the wellbore, preventing fluids such as oil, gas, and water from migrating along the outside of the casing, protecting the reservoir and surrounding environment, and providing structural support to the wellbore. During drilling and completion operations, cement slurry is injected into the annular space between the borehole and casing, where it hardens to form a cement sheath. This sheath needs to exhibit good mechanical strength, adhesion, and low permeability to effectively withstand various factors like external pressure changes, temperature fluctuations, and chemical corrosion. Additionally, the cement sheath must maintain stable sealing performance throughout the well’s lifespan to prevent any fluid leakage. This stability is vital for protecting the subsurface reservoirs and ensuring long-term well productivity. In petroleum engineering, the permeability of the cement sheath is critically important, as it directly affects the sealing effectiveness of the well. If the cement sheath's permeability increases, fluids may penetrate through it, potentially leading to fluid migration between the reservoir and surrounding formations. This can result in contamination between formations and could even harm the surrounding ecological environment.[1-2].

2. Improve the typos errors in whole manuscript. For now:

“CO2” in line 75; “..” in line 182; “(3.9 \\times 10{-9})” in line 190; Check line 206;

“1.54×10-2” in line 279; “from 1 mm to 1.19×10-1 m3/d” in line 417.

Dear reviewer

I sincerely apologize for our oversight. We have carefully corrected these errors and conducted a thorough review of the entire paper.

3. There are many abbreviations used in this paper. It is advised to include a table for all abbreviations used in the paper.

Dear reviewer

There are only two abbreviations for the full text, we think it is not necessary to establish abbreviations, and please reconsider.

sustained casing pressure (SCP)

casing-cement interface (CCI)

4. It is also advised to provide citations for all governing equations and utilised formulas. Also, use proper syntax (comma or full stop) at the end of displayed equations

Dear reviewer

We have added references and added proper syntax.

5. Give the references for the chosen numerical range of parameter values of emerging parameters.

Dear reviewer

We have given the references for the chosen numerical range of parameter values of emerging parameters.

6. Try to exclude access use of indent in each paragraph.

Dear reviewer

We have reduced paragraph inentation.

7. For instance, address the following articles in context of flow through porous media:

a) https://doi.org/10.1016/j.chaos.2024.114961

b) https://doi.org/10.1002/zamm.202200047

c) https://doi.org/10.1016/j.chaos.2024.114726

d) https://doi.org/10.1016/j.cjph.2024.01.017

b) https://doi.org/10.1016/j.cjph.2024.03.046

Dear reviewer

We have added references to the following article.

a) https://doi.org/10.1016/j.chaos.2024.114961

28.Yadav P K, Yadav N. Impact of heat and mass transfer on the magnetohydrodynamic two-phase flow of couple stress fluids through a porous walled curved channel using Homotopy Analysis Method[J]. Chaos, Solitons & Fractals, 2024, 183: 114961.

b) https://doi.org/10.1002/zamm.202200047

29.Yadav P K, Verma A K. Analysis of two non‐miscible electrically conducting micropolar fluid flow through an inclined porous channel: Influence of magnetic field[J]. ZAMM‐Journal of Applied Mathematics and Mechanics/Zeitschrift für Angewandte Mathematik und Mechanik, 2023, 103(1): e202200047.

Reviewer #2: This reviewer carefully read the manuscript and found it to be well-written. The study focused on the finite element methods to develop a simulation model for cement sheath seepage. It analyzes parameters influencing seepage flow, cement sheath length, permeability, gap structure, pressure difference, and fluid characteristics. However, before accepting the manuscript, the corresponding author/s should make some minor corrections pointed out in the manuscript, including typo errors.

Dear reviewer

We have conducted a thorough review of the entire text, including checks for grammatical errors and typos.

---

## [Decision Letter · Decision Letter 1]

4 Dec 2024

Analysis of key parameters influencing the permeability of cement sheath based on multiphysical fields

PONE-D-24-39005R1

Dear Dr.chengyu xia,

We’re pleased to inform you that your manuscript has been judged scientifically suitable for publication and will be formally accepted for publication once it meets all outstanding technical requirements.

Kind regards,

Ajaya Bhattarai

Academic Editor

PLOS ONE

Additional Editor Comments (optional):

Reviewers' comments:

Reviewer's Responses to Questions

**Comments to the Author**

1. If the authors have adequately addressed your comments raised in a previous round of review and you feel that this manuscript is now acceptable for publication, you may indicate that here to bypass the “Comments to the Author” section, enter your conflict of interest statement in the “Confidential to Editor” section, and submit your "Accept" recommendation.

Reviewer #1: All comments have been addressed

Reviewer #2: All comments have been addressed

2. Is the manuscript technically sound, and do the data support the conclusions?

Reviewer #1: Yes

Reviewer #2: Yes

3. Has the statistical analysis been performed appropriately and rigorously? 

Reviewer #1: Yes

Reviewer #2: N/A

4. Have the authors made all data underlying the findings in their manuscript fully available?

Reviewer #1: Yes

Reviewer #2: Yes

5. Is the manuscript presented in an intelligible fashion and written in standard English?

Reviewer #1: Yes

Reviewer #2: Yes

6. Review Comments to the Author

Reviewer #1: Title: Analysis of key parameters influencing the permeability of cement sheath based

on multiphysical fields

Comments:

The authors have improved the manuscript. Now, it can be accepted

Reviewer #2: The revised manuscript incorporated the comments and typo corrections as pointed out in the first comments.

7. PLOS authors have the option to publish the peer review history of their article (what does this mean?). If published, this will include your full peer review and any attached files.

Reviewer #1: **Yes: **Dr. Pramod Kumar Yadav

Reviewer #2: No

---

## [Editor Report · Acceptance letter]

10 Dec 2024

PONE-D-24-39005R1 

PLOS ONE

Dear Dr. xia, 

I'm pleased to inform you that your manuscript has been deemed suitable for publication in PLOS ONE. Congratulations! Your manuscript is now being handed over to our production team.

Kind regards, 

on behalf of

Dr. Ajaya Bhattarai 

Academic Editor

PLOS ONE